# Influence of Walking, Manual Techniques, and Elastic Resistance Exercise on Shoulder Posture in Healthy Elderly Individuals

**DOI:** 10.3390/geriatrics9050128

**Published:** 2024-10-04

**Authors:** Klára Novotová, Dagmar Pavlů

**Affiliations:** Faculty of Physical Education and Sports, Charles University, CZ-162 52 Prague, Czech Republic; novotova.klara@ftvs.cuni.cz

**Keywords:** walking, resistance training, scapular position, acromion-wall distance, elderly, healthy

## Abstract

In this study, we investigated the effect of regular walking and its combination with manual techniques/resistance exercise. The position of the shoulder girdle was assessed using the acromion-wall distance (AD). The intervention took place twice a week for 4 weeks. A total of 88 seniors over the age of 60 successfully completed the study. The results showed a statistically significant improvement of AD in the left shoulder within the group that underwent walking combined with resistance exercise. The remaining groups did not show any statistically significant change in AD. Background: The world population is rapidly aging; therefore, it is necessary to respond to this challenge in time. One of the typical involutional signs of old age is a hunched posture combined with a forward position of the shoulder girdle. This posture negatively impacts various bodily functions, postural stability, and strain on the musculoskeletal system. Objectives: We aimed to evaluate the effect of walking and walking combined with manual therapy/resistance exercise on scapular positioning in healthy elderly individuals. Methods: Participants of experimental groups underwent a 4-week training session that involved walking and manual techniques/resistance training applied 2x/week. Participants of the control group maintained their usual daily habits. Results: Our results showed statistically significant improvement in scapular positioning of the left shoulder in participants who underwent regular walking combined with resistance exercise training. Conclusions: These results suggest that regular walking combined with resistance training, when properly dosed, may beneficially influence scapular positioning in healthy elderly individuals.

## 1. Introduction

The world faces major challenges as the population ages faster and to a greater extent than ever before. It is therefore crucial to maintain the overall health of elderly people to delay their care dependency as long as possible [1].

There is no universally agreed-upon age at which a person is considered old. Various sources differ on the definition of old age. Some organizations define the older population as people aged 65 and above [2]. However, for the purpose of this study, we defined elderly people as those above 60 years of age, which aligns well with definitions provided by both the World Health Organization and the United Nations. Both organizations define older persons as people above 60 years of age, as it is considered globally as a retirement age, and people older than that are by some considered economically inactive [3,4].

A hunched (kyphotic) posture is typical for older age, affecting 20–40% of the population over 60 years of age [4]. It usually results from muscle weakness, spinal degeneration, and osteoporotic processes [5].

For the hunched posture, a typical forward spine curvature is present, causing the shoulders and upper back to round forward. This forwarding usually results in scapular protraction, where the shoulder blades move away from the spine, tilting outward and shifting forward [6,7]. However, scapular protraction does not occur only in older people with a hunched posture. Scapular protraction is a common condition in modern society and is associated with other factors besides kyphosis, such as shoulder instability [8], muscle imbalances [9], or repetitive stress injuries [10]. In this study, we aimed to include participants from the general older population in which musculoskeletal factors impact shoulder positioning in terms of scapular protraction, whether kyphosis is present or not. Scapular protraction is then linked to forward head posture [11], impaired lung function [11], increased risk of falls [12], gastrointestinal dysfunction, and overall quality of life [4]. The rehabilitation techniques for scapular protraction management are an area of interest among researchers, and various physiotherapeutic interventions are used in clinical practice to beneficially influence scapular protractions. These techniques include strengthening exercises [13], stretching [14], postural training [15], ergonomic adjustments [15], and body awareness practice [14]. Most of these interventions were performed by a physical therapist; however, we preferred the participants to self-administer all exercises under supervision.

There is limited research on a direct link between scapular protraction and walking interventions. However, it has been proven that walking can improve body alignment, potentially reducing the tendency for scapular protraction [16,17]. When it comes to kyphotic posture and walking intervention, results are not universally conclusive. Studies have shown both the existence of a beneficial effect of walking [18] and its modifications on kyphotic posture as well as its absence [19]. Studies on manual therapy and manipulative techniques explore their impact on scapular protraction. These studies showed a beneficial effect on scapular protraction, and the interventions performed include scapular mobilizations [20], soft tissue work [21,22], and muscle energy techniques [20]. Few studies aimed to examine the effect of manual techniques on kyphotic posture in older adults. These studies suggest that manual techniques can attenuate thoracic kyphosis in patients suffering from osteoporosis [23] and vertebral osteoporotic fractures [24]. In these studies, the common dosage involved administering manual techniques 1–2 times per week, for a total of 6–12 weeks, integrating manual therapy with various subsequent exercises, usually resistance band exercises focused on proper scapular alignment.

Research on resistance exercise provides evidence of its beneficial influence on scapular positioning [25,26] and kyphosis reduction [27,28]. By targeting specific muscle groups, it can correct postural deviations and contribute to the physiological alignment of bodily segments. These interventions included exercises aimed at rotator cuff muscles by performing rows, scapular retraction, and external rotation using elastic resistance bands [25]. In another study, shoulder blade squeezes, prone Y/T raises, and scapular push-ups using resistance bands were performed [26]. To reduce kyphosis using resistance machines or bands, lat pulldowns [27], chest flys [27], back extensions [27,28], seated rows [28], and reverse flys were performed [28]. In general, the duration of these exercise programs ranged from 6 to 12 weeks, with exercise sessions performed 2–3 times a week for 30–60 min each.

The purpose of this study was to evaluate and compare the effect of walking and walking combined with manual techniques/elastic resistance exercise on the forward shoulder posture evaluated by acromial distance in healthy elderly individuals. The individual favorable effect of manual techniques/elastic resistance exercise on kyphotic posture has already been discussed. The present study combines walking and manual techniques/elastic resistance exercises for potentially increased efficiency of the interventions.

Another aim of this study was to create a basis for the design of a simple preventive exercise program for the general senior population. For the future, we hope that this program could be adhered to individually in order to adjust body alignment and thereby delay the onset of the above-described difficulties related to poor body alignment.

## 2. Materials and Methods

### 2.1. Participants

A total of 88 participants (75 female, 13 male) aged 70.59 ± 5.42 years (mean ± SD) completed the study (Table 1).

### 2.2. Inclusion Criteria

Participants had to meet the following criteria: aged 60–80 years old, able to walk without gait aids for at least 20 min, Charlson comorbidity index (CCI) value of 3 or less, absence of serious cardiorespiratory disease, nonsmokers for at least the last 6 months, absence of contraindications to the performance of manual techniques in the chest area, absence of cognitive deficit, and ability to provide informed and voluntary written consent to participate in the research.

### 2.3. Exclusion Criteria

The exclusion criteria were set as follows: age outside the range of 60–80 years, inability to walk unaided and provide informed written consent to participate in research, CCI value above 3, existence of serious cardiorespiratory disease, active smoking, smoking in the last 6 months, existence of contraindications to the performance of manual techniques to the chest area, and Mini Mental State Exam score outside the range of 24–30. The participants meeting any of these criteria were excluded from the study, while the remaining ones were assigned to the experimental/control group.

### 2.4. Randomization

The process of randomization was held as follows: 96 sealed envelopes were prepared. A total of 24 contained the number “1” representing the first experimental group, 24 contained the number “2” representing the second experimental group, 24 contained the number “3” representing the third experimental group, and 24 contained the number “4” representing the control group. Each participant randomly selected a sealed envelope from the set of prepared envelopes to be assigned to a specific group.

### 2.5. Dropout Rates

A total of 8 participants dropped out of the study. The data of these participants were not included in the study.

### 2.6. Experimental Design

The experimental intervention lasted for 4 weeks. There were 3 experimental groups in this study labeled “1–3”. All 3 groups participated in training sessions held under supervision 2x/week. The control group was asked to maintain their regular routine and not to change it in any way.

### 2.7. Sample Characteristics

All participants were evaluated and interviewed during a pre-intervention assessment. Both men and women who met the inclusion criteria enrolled in this study. The following table summarizes the basic characteristics of participants.

### 2.8. Intervention

Group 1 (walking group) underwent regular walking alone, 40 min per session. The walking training took place in an indoor gym with a flat, dry surface. The walking intensity was determined using the Borg scale of perceived exertion. The intensity progressively increased during each training session.

Group 2 (manual techniques + walking) underwent 20 min of self-administered manual techniques focused on the thoracic and cervical region followed by 40 min of walking identical to Group 1. The manual techniques included massage of soft tissues in the thoracic and neck region, stretching of thoracic and cervical muscles, as well as mobilization of joints in the mentioned bodily regions. Participants were also educated about diaphragmatic breathing. The manual techniques training protocol was designed based on studies dealing with manual techniques and their impact on posture and overall health in the elderly [24,25].

The training protocol for manual therapy follows:

1. Training of diaphragmatic breathing lying in the supine position (1 min)

2. Mobilization of the cervical spine by active prolongment and traction of the cervical spine in the supine position (15 rep)

3. Mobilization of the shoulder girdle by circular motion of the shoulders in the standing position (0.5 min)

4. Massage of intercostal muscles in the region parallel to the sternum/thoracic vertebrae in the standing position (2 min)

5. Mobilization of the upper cervical spine and cervico-cranial joints by active movement in protraction/retraction of the head in the standing position (15 rep)

6. Self-massage of sternocleidomastoid muscles in the standing position (4 min)

7. Self-massage of trapezoid muscles in the standing position (4 min)

8. Training of diaphragmatic breathing in the standing position (1 min)

9. Mobilization of the spine in lateroflexion and stretching of lateral trunk muscles combined with diaphragmatic breathing (3 min)

10. Mobilization of shoulder blades by active movement of protraction and retraction combined with diaphragmatic breathing (15 rep)

11. Same as step 1 (1 min)

Group 3 (elastic band resistance training + walking) underwent 20 min of elastic band resistance training followed by 40 min of walking identical to Group 1. The elastic band resistance included a series of exercises focused on the correction of the kyphotic posture. The participants performed a total of 3 exercises (Figure 1, Figure 2 and Figure 3) in 3 series and 10 repetitions within each series for both shoulders. There was always 1 min of rest between individual series. The number of repetitions gradually increased up to 12 between training sessions. For the exercises, an elastic resistance band (Sanctband band Orange (soft) and Lime green (medium), the length 3.5 m) commonly used in training interventions was used [29]. All resistance exercises were designed based on available knowledge [29]. The 3 exercises were performed in the seated position and included:

Group 4 (control group) was asked to maintain their regular routine and not to change it in any way.

### 2.9. Outcome Measure

To evaluate the position of the shoulder girdle over time, acromion distance was determined. Acromion distance (AD) is the horizontal distance between the wall and the posterior border of the scapular acromion process. It is used to determine scapular dysfunction in terms of pectoralis minor muscle shortening. This muscle, when shortened, often causes an increased internal rotation and anterior tilting of the scapula and is related to rounded or forward shoulder posture [30]. Acromion distance measurement (ADM) in the upright position has been approved both for validity and reliability. For ADM validity while standing, a moderate correlation with radiographic imaging has been proven [31]. For ADM reliability while standing, acceptable interrater [32] and adequate intra rater [33] reliability for clinical practice has been demonstrated. The ADM was performed with the participant standing with the back facing the wall. The assessor then instructed the participant to stand relaxed while putting the feet and thorax against the wall. The assessor measured the horizontal distance between the most posterior part of the acromion and the wall for both shoulders using a sliding caliper.

### 2.10. Statistical Analysis

All statistical analyses were executed in R (4.3.2) and the results were evaluated to identify any significant results among the groups. Descriptive statistics, including mean, standard deviation, minimum, maximum, and 1st and 3rd quartiles, were used to summarize the measured outcomes. The normality of data was assessed using the Shapiro–Wilk test. Given that the assumption of normality was not met, we employed the Kruskal–Wallis test to evaluate whether there were statistically significant differences in the distribution of the variable of interest (AD) across the different groups. The Wilcox signed-rank test was used to compare AD within the same group before and after treatment. Following a significant Kruskal–Wallis test, we conducted Dunn’s test with Bonferroni correction to assess pairwise differences among the groups. The Dunn’s test was selected as a non-parametric post-hoc test to address multiple comparisons while controlling for Type I error. In this study, a *p*-value of less than 0.05 was considered statistically significant. The effect size (r) for the Wilcoxon signed-rank test was computed.

## 3. Results

The results are shown in Table 2 for the AD measured on the right shoulder before and after intervention. Table 3 shows AD measured on the left shoulder.

Based on the results provided in Table 2, we can summarize that AD (right shoulder) slightly decreased in all four groups. This change was not significant in any of the groups.

The Dunn’s test with Bonferroni correction was performed to identify significant pairwise differences among the four groups for each variable of interest (AD before and after treatment in both shoulders). There were no significant differences in distributions of AD before treatment in the right shoulder, AD after treatment in the right shoulder, and AD before treatment in the left shoulder across the four groups. The comparison of AD after treatment across the four groups showed no significant differences as well, except for the comparison of group 3 (elastic resistance training) and group 4 (control group), with an unadjusted *p*-value of 0.0091. However, after the Bonferroni correction, the adjusted *p*-value was slightly above the typical significance threshold (*p* = 0.0546). In summary, none of these pairwise comparisons yielded statistically significant differences after being adjusted for multiple comparisons using the Bonferroni method.

## 4. Discussion

This preliminary study aimed to investigate the effect of walking and walking combined with manual techniques/elastic resistance exercise on shoulder position in healthy elderly individuals. To our knowledge, the available literature usually analyzed individual interventions separately or with a different target population. These studies support the assumption that elastic resistance exercises are effective in improving shoulder posture and performance [34]. A systematic review has highlighted the beneficial effects of elastic band resistance training on upper limb posture, flexibility, strength, and endurance [35]. Another study states that the integration of walking with other forms of treatment, such as manual techniques and elastic resistance, positively impacts shoulder posture and stability [36].

There is an intense debate about the role of the upper limbs in walking. Most researchers accept the fact that there is a contralateral swing to the lower limbs, but there is a disagreement as to why it occurs. Some believe that this contralateral swing of the upper limbs during walking is actively generated by the body, while others argue that this is merely a passive response of the body to the rotation generated by the lower limbs during the stride cycle [37]. Research has shown that walking has a direct impact on shoulder positioning. The rhythmic nature of walking influences shoulder positioning in terms of scapular movement [38]. Walking engages upper body muscles in both eccentric and concentric modes. Regular engagement of these muscles during walking can improve their strength and coordination, leading to improved scapular stability and positioning [39]. Over time, balanced muscle activity helps prevent excessive scapular protraction, reducing the stress on the shoulder joints [39]. Scapular positioning can be also improved through the engagement of the core muscles during walking, especially the abdominals and lower back muscles, thus contributing to physiological trunk alignment and reducing the tendency for scapular protraction or winging [39]. Regular walking reinforces correct movement patterns through repetition. The nervous system becomes more efficient in managing and coordinating muscle activity regarding scapular positioning [40]. We can summarize that regular walking has the potential to reduce shoulder dysfunction and improve overall shoulder health.

In this study, the participants of three intervention groups performed 40 min of regular walking (2x/week, 4 weeks total). The dosage of walking was derived from WHO’s guidelines on physical activity for adults aged 65 and older [1]. These recommendations specify the amount of physical activity older people should engage in on a regular basis to prevent involutional health conditions. It is strongly recommended for older adults to do at least 150–300 min of moderate-intensity activity or at least 75–150 min of vigorous-intensity activity or an equivalent combination of the two throughout the week. The participants of Group 1 (walking group) gradually performed 80 min of vigorous-intensity walking (Borg RPE scale 15–16) per week. Participants of Group 2 and Group 3 also performed an additional activity for potentially increased efficacy. Our results showed no significant improvement of AD within or between Group 1 with small effect sizes and other groups, suggesting that 40 min (2x/week, 4 weeks total) of regular vigorous walking had no significant effect on shoulder position. There are not many studies focusing on regular walking as a form of training and its direct impact on shoulder position; therefore, it is not possible to compare our results with other studies. Based on a slight improvement in AD in Group 1 after the intervention, we hypothesize that the form of exercise or exercise dosage in this group was not sufficient to improve AD, as the biomechanical basis for possible improvement exists and has been described above. While regular walking provides numerous health benefits, we hypothesize, that in order to successfully improve scapular positioning, walking intervention should be performed in some sort of modification, such as Nordic walking, which enables the engagement of upper limbs more efficiently during each stride [41]. In our future research, we will consider examining Nordic walking instead of regular walking in the elderly to improve scapular protraction.

Manual techniques are methods designed to assess, diagnose, and treat various conditions. These techniques involve the use of the therapist’s hands to mobilize joints, manipulate soft tissues, improve movement, promote circulation, relieve pain, and enhance overall health [42]. Manual techniques in physiotherapy work on a mechanical, neurological, and physiological basis. The mechanical aspect involves the application of force to joints or soft tissues. The aim of these techniques is to improve the joint range, reduce stiffness, and correct joint alignment [43]. Soft tissue mobilization involves the manipulation of muscles, fascia, and other soft tissues to break down adhesions, promote circulation, improve tissue elasticity, and promote fiber alignment within tissues [44]. The neurological effect of manual techniques lies in pain modulation, muscle relaxation, and proprioceptive reeducation [42]. From a physiological point of view, soft tissue techniques improve circulation with tissues as well as lymphatic drainage and tissue remodeling. The beneficial psychological effects of manual techniques should not be neglected [43]. We can conclude that through mechanical adjustment, stimulation of the nervous system, and physiological changes in target tissues, manual techniques help restore the normal function of these tissues.

Participants in Group 2 (manual techniques) underwent an exercise intervention of walking combined with manual techniques. The exercise session proceeded as follows: participants performed a series of massage palpations designed to release tension from cervical and thoracic tissues; mobilize joints located in the cervical and thoracic region; and stretch trapeze, sternocleidomastoid, and pectoral muscles. Participants were instructed on physiological breathing mechanics and patterns, which they then practiced in various positions. The entire exercise unit took place under the supervision of a physiotherapist. The exercise session lasted 60 min: during the first part of the session (20 min) participants performed self-administered manual therapy directly followed by a 40-min walking session identical to Group 1.

Our results show a slight decrease in AD in Group 2 with small effect sizes; however, this decrease was not statistically significant. Studies examining the effect of manual techniques on shoulder position in patients without pain or impingement syndrome are relatively rare. These studies suggest that manual techniques are efficient in improving shoulder positioning [45,46], scapular kinematics [47], shoulder mobility [45], and perceived pain levels [46], mostly in patients suffering from impingement syndrome. The lack of significant changes in AD in our study can be explained by the lack of additional exercise. Various studies have demonstrated that in order to improve scapular positioning and other beneficial effects, manual techniques should be combined with some sort of therapeutic exercises [46,48]. The underlying mechanism lies in the synergistic effect of manual techniques and subsequent exercises. Manual techniques immediately normalize the tension in soft tissues, improve joint mobility, and reset the neuromuscular system, making the patient more receptive to subsequent exercise. When combined with active exercise, there is a possibility of addressing both active (muscle endurance, muscle strength, neuromuscular control) and passive (tissue flexibility, joint mobility) components of the musculoskeletal system, making the intervention more efficient than the individual interventions separately [46,48,49]. We hypothesize that, in general, manual techniques and mobilizations alone are not sufficient for long-term correction of scapular protraction and body alignment. While these techniques can help with soft tissue and joint flexibility, they do not target muscle strength in the muscles responsible for stabilizing the scapula [49]. Since scapular protraction usually results from muscle imbalances, strengthening these muscles is crucial for long-term results [32]. In our study, manual techniques combined with walking were not sufficient to significantly improve scapular protraction. This is caused by their lack of ability to strengthen crucial muscles and provide the muscle activation required to address the underlying muscle imbalances of scapular protraction. To conclude, better outcomes in scapular positioning can be ensured by combining manual techniques with subsequent exercises.

Resistance exercise is a form of physical activity that uses external resistance in order to improve muscle strength, endurance, and size. The external resistance can originate from various sources, e.g., free weights, resistance bands, weight machines, or body weight [50]. The relationship between resistance exercise and shoulder posture has been discussed in many previous studies. Results indicate that resistance training can improve both scapular positioning and shoulder posture [51,52,53]. Resistance exercises improve scapular positioning through several mechanisms contributing to physiological alignment, stability, and function of the shoulder girdle. Resistance training targets various muscles located in the scapular region, and by strengthening these muscles, it allows proper stabilization of the shoulder joint and, thus correct scapular positioning [51]. Another mechanism by which resistance training affects the position of the scapula is improving muscle imbalances, enhancing postural control, and increasing proprioception [54,55]. An improvement in proprioception helps in maintaining proper shoulder positioning during different daily activities [51]. By enhancing muscle strength, muscle endurance, joint stability, and proprioception, resistance training plays an essential role in proper shoulder and scapular positioning.

Participants in Group 3 (resistance training) performed a series of exercises with elastic resistance bands. Resistance training was performed in a seated position and focused on posture correction. Exercises were completed with controlled breathing, and each exercise was repeated in several sets. The entire exercise unit took place under the supervision of a physiotherapist. The training session lasted 60 min: during the first part of the session (20 min), participants performed elastic resistance exercises directly followed by a 40-min walking session identical to Group 1.

Our results showed a statistically significant decrease in AD for the left shoulder (*p* = 0.001) with a large effect size of r = 0.74, while there was no significant change for the right shoulder. Our findings were surprising, as similar results are seen in literature focusing on resistance exercises combined with stretching usually after six weeks of training [56].

We hypothesize that the absence of effect for the right shoulder may stem from an insufficient dosage of exercise. The unilateral improvement in scapular protraction in Group 3 could be explained by several factors related to the combination of walking and resistance exercises, such as asymmetry in muscle strength or use. These participants may have had pre-existing biomechanical imbalances where the left scapula was more flexible or weaker, making it easier to show visible improvement after only four weeks of training. If participants had a tendency to favor the left shoulder during their exercises, these muscles may have been more efficiently engaged, leading to a greater improvement in scapular positioning compared to the right side. As the majority of participants were right-handed, overuse of the right hand could lead to more complex postural changes on the right shoulder, which could be corrected with a more intensive training plan. We assume that these pre-existing asymmetries were beyond our abilities to detect during our assessments, as the baseline AD in Group 3 was similar for both shoulders: 9.03 cm (1.32) for the right shoulder and 9.05 cm (1.37) for the left shoulder. Another possible explanation for these results may be an unequal distribution of study participants in terms of AD baseline value, as the baseline values of AD for both shoulders are higher in Group 3 compared to the other groups. Participants may have also been unequally distributed into groups by activities they usually participate in, which may have influenced our findings.

The evidence combining specifically resistance training with walking is more indirect. To our knowledge, there are not many studies isolating the effect of resistance exercise combined with walking. One study suggests that comprehensive rehabilitation programs, including resistance exercise training, flexibility, and aerobic training (such as walking) can lead to better scapular positioning [57]. This study emphasized the holistic benefits of both types of activities combined. Other studies suggest that a combination of resistance exercise with other general physical activities may also be a more efficient approach to improve scapular positioning than resistance training alone [58,59]. We assume that further research in this area would bring clearer results and clinical recommendations.

The results obtained in Group 4 (control group) showed a slight decrease in AD for the right shoulder with a small effect size. AD for the left shoulder increased (*p* = 0.11) with a medium effect size of 0.51. This can be explained by a high variability in data. To summarize, the control group did not show any significant change in AD for either shoulder.

This study has several limitations. The sample size in this study may be too small, thus reducing the statistical power of the study. This sample also represents elderly adults who are relatively fit and healthy, limiting the generalizability of the results to the entire adult population. Participants volunteered to enter the study, which can potentially skew the selected sample. Based on the randomization process in this study, there could be a non-homogenous distribution of participants into individual groups. The effect of manual techniques and resistance exercise was not investigated separately, only in combination with walking. All these limitations should be taken into consideration before formulating any conclusions.

## 5. Conclusions

In this study, we aimed to examine the effect of regular walking and its combination with manual therapy/resistance exercises on scapular positioning in healthy elderly individuals. Participants of the experimental groups underwent training sessions 2x/week for a total of 4 weeks. Our results showed that after the application of targeted walking training in the senior population for 60 min twice a week in combination with resistance exercises using elastic resistance on the trunk and upper limbs, there was a significant improvement in the position of the arm girdle on the left (non-dominant) upper limb. However, the underlying biomechanical mechanisms for this improvement should be examined in order to provide clear explanations. Walking and its combination with manual techniques did not show any significant results in scapular positioning. Further research on postural correction using walking combined with subsequent exercise in the elderly is needed to formulate clear results and make clinical recommendations. However, at this point, we can say that the use of guided walking and simple elastic resistance exercise appears to be a simple procedure that can help improve the position of the girdle.

## Figures and Tables

**Figure 1 geriatrics-09-00128-f001:**
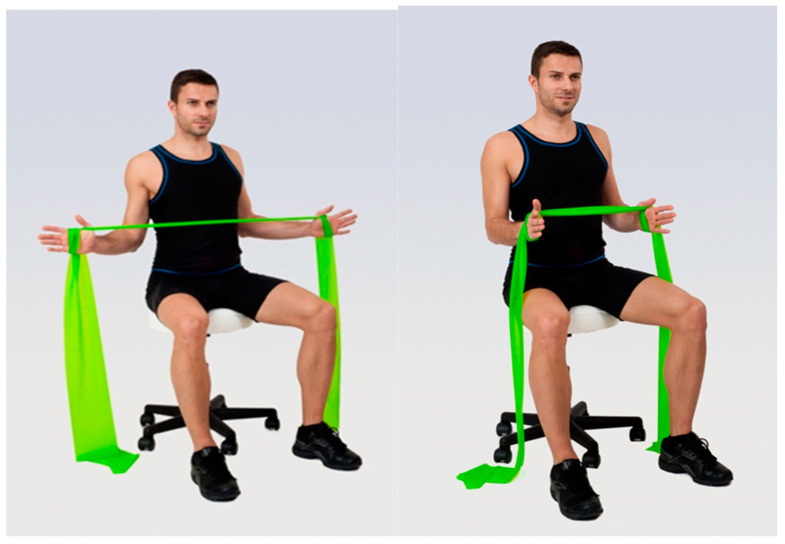
Illustrative picture of exercise 1—the exercise supports external rotation in the shoulder joints and straightening of the trunk: in the first phase of the exercise, external rotation is performed in the shoulder joints against the resistance of the elastic band (concentric contraction of the external rotators of the shoulder) and at the same time the upright posture of the trunk is emphasized; in the second phase, the exerciser slows the movement into internal rotation, into which it is moved by an elastic band (eccentric contraction of the internal rotators of the shoulder). The activity of the muscles of the internal rotators of the shoulder joints is reduced by the given exercise within the principle of reciprocal inhibition.

**Figure 2 geriatrics-09-00128-f002:**
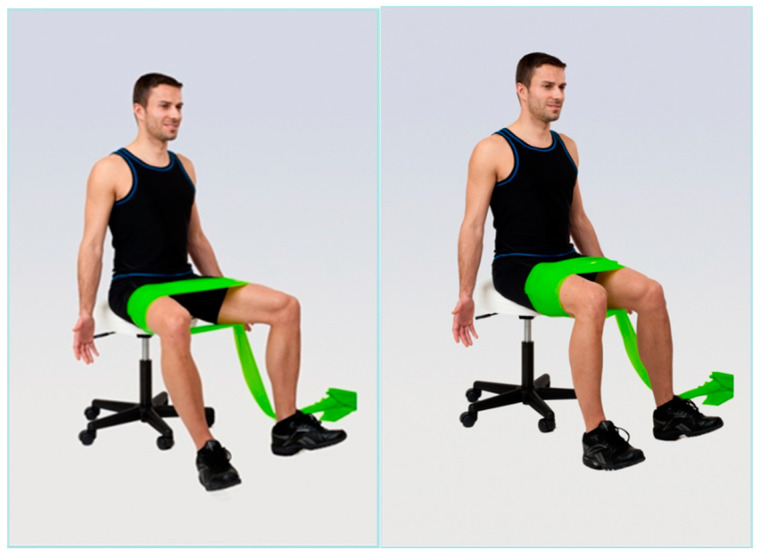
Illustrative picture of exercise 2—the exercise supports the activity of the external rotators of the hip joints and the straightening of the trunk: in the first phase of the exercise, a concentric contraction of the external rotators of the hip joint is performed against the resistance of the elastic band; in the second phase, an eccentric contraction of the external rotators of the hip joint, while simultaneously controlling the straightening of the trunk, the exercise is combined by actively turning the arms into external rotation.

**Figure 3 geriatrics-09-00128-f003:**
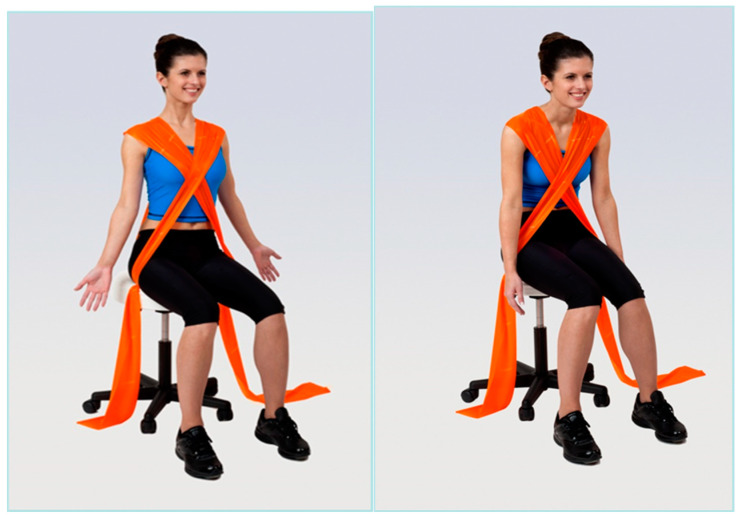
Illustrative picture of exercise 3—the exercise supports an upright posture and opening of the ventral side of the chest: in the first phase of the exercise, the exerciser straightens the trunk (concentric contraction of the muscles that straighten the trunk) against the resistance of the elastic band; in the second phase of the exercise, the exerciser slows down the movement until the trunk bends into which the elastic band pulls it (eccentric contraction).

**Table 1 geriatrics-09-00128-t001:** Baseline characteristics of participants.

Baseline Characteristics	Total, *n* = 88
Age, years, (mean ± SD) Sex, female, *n* (%) Sex, male, *n* (%) 13 (15) Education, *n* (%) Primary education High school/vocational school University degree	70.59 ± 5.42 75 (85) 1 (1.1) 44 (50) 43 (48.9)
BMI (kg/m^2^), *n* (%) Underweight (<18.5) Healthy weight (18.5–25) Overweight (25–30) Obesity (>30) Weight (kg), (mean ± SD) Height (m), (mean ± SD)	0 50 (56.8) 24 (27.3) 14 (15.9) 70.52 (14.1) 1.66 (0.9)

**Table 2 geriatrics-09-00128-t002:** Acromion distance (cm)–right shoulder. Before and after intervention.

		GROUP 1	GROUP 2	GROUP 3	GROUP 4
AD (cm) right side before intervention	Mean (SD)	8.61 (1.61)	8.45 (1.6)	9.03 (1.32)	8.37 (1.5)
	Min.	5.5	4.5	6.5	5.9
	1st Q	7.6	7.9	8.1	7.2
	3rd Q	10.08	9.5	9.7	9.4
	Max.	11.5	10.9	12.5	11.8
AD (cm) right side after intervention	Mean (SD)	8.31 (1.47)	8.19 (1.67)	8.83 (1.44)	7.91 (1.62)
	Min.	5.1	5	6.5	5.2
	1st Q	7.12	7.12	7.9	7
	3rd Q	9.45	9.15	9.6	8.8
	Max.	10.9	11	12.5	12
	*p*-value	0.59	0.54	0.58	0.28
effect size	r	−0.12	0.13	−0.04	0.34

**Table 3 geriatrics-09-00128-t003:** Acromion distance (cm)–left shoulder. Before and after intervention.

		GROUP 1	GROUP 2	GROUP 3	GROUP 4
AD (cm) left side before intervention	Mean (SD)	8.5 (1.75)	8.43 (1.68)	9.05 (1.37)	8.42 (1.14)
	Min.	5.7	4.5	6.3	6
	1st Q	7.12	7.4	8.3	8
	3rd Q	10.3	9.87	9.8	8.9
	Max.	11.8	10.5	12	12
AD (cm) left side after intervention	Mean (SD)	8.14 (1.6)	8.04 (1.52)	7.74 (1.45)	8.99 (1.36)
	Min.	5.3	4.5	4.2	6.7
	1st Q	7.05	7.15	7.35	8.1
	3rd Q	8.45	9	8.25	9.7
	Max.	11.5	10.6	10	11.5
	*p*-value	0.41	0.34	**0.001**	0.11
effect size	r	0.08	0.38	0.74	0.51

## Data Availability

In case of interest, all materials and datasets can be provided by either of the authors.

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
