# Peer review of "Influence of Walking, Manual Techniques, and Elastic Resistance Exercise on Shoulder Posture in Healthy Elderly Individuals"

_geriatrics, 2024, doi:10.3390/geriatrics9050128_

Round 1

Reviewer 1 Report

Comments and Suggestions for Authors

The study on the shoulder posture of the elderly, which is rapidly increasing worldwide, is very interesting and I think it is a study appropriate for the times.

In addition, after reviewing this paper, I would like to ask you some questions and requests.

1.

The subjects of the experiment were elderly. Please provide the basis for defining the elderly age as 60 years or older. It is generally defined as 65 years or older.

2.

In the introduction, why is there so little explanation about shoulder posture deformation in the elderly, and instead so much explanation about kyphosis?

Did you want to study shoulder deformation in patients with kyphosis?

Please describe the subjects clearly. Are the subjects elderly with kyphosis and shoulder deformation?

3.

Please add a description of the massage and stretching techniques applied to Group 2 to the text.

4.

I would appreciate it if you could insert a photo of the band resistance exercise applied to Group 3 into the text.

Author Response

See attachement

Reviewer 2 Report

Comments and Suggestions for Authors

Overview.

This is an experimental study of the impact of walking and manual/elastic resistance exercise on shoulder posture in healthy elderly. Hope Behavioral InterventionH This is a well-designed and well written manuscript.

Introduction The introduction provides a strong argument for the need to determine which interventions may be best to ultimately reduce the negative impact of kyphosis on older adults.  This section concludes with a clear purpose statement.

Materials and Methods: .

Participants: Suggest “A total of __ participants ages [provide the range], averaged age 70.59 (s.d. 5.42 years) were recruited over [time frame] from [where?]  It is usual to be clear if the participants came from independent living situations or not and how long it took to recruit them.  You do not need to say the actual place but may say for example, “a large urban medical center” or rural club for older adults, etc.  The reason the time is given is to help the reader know if history could have had an impact (such the pandemic).

Also please address the discrepancy about the total number recruited, enrolled and completing the study.  You list 88 as the n, but then say in line 84, that 8 participants “dropped out”.  Is your final n 96 or 80?  Please correct throughout the tables and paper on this point.

Inclusion and exclusion criteria are clear.

Statistics including the various corrections are appropriate.

Results:  This section is organized well with all Tables (1-3) providing details summarized in the body of the text.

Discussion: The discussion section is particularly well done and the reader appreciates the detailed descriptions of the interventions used in each group including results for the control group.

The limitations related to sample size, multiple tests and the possibility that true randomization may not have occurred are appropriate.

Conclusions: Brief and logical, definitely more research is needed.

Minor editing:

Line: 74 change “expelled” to “excluded”

Line 118 change “ seating” to “seated”

Line “40 change "proven” to “demonstrated”

Line 183 omit words “so far” 

Line 196 omit word “anyway”

Line 197 “influence” should be “influences”

Line 260 change “proven” to “demonstrated”

Line 262 add “the” to the neuromuscular system, make the patient more receptive…”

Line 293 add “the to “…the left shoulder”

Line 294 add “the” to “for the right..”

Line 296 add “the to “…the right shoulder"

Line 306 omit “specifically”

Line 312 change “suggests” to “suggest” no s

Line 313 restate phrase “…activities may also be more…”

Line 339 omit word “have”

Line 342 add “the” so reads “The influence of walking…”

Line 343 change “however” to “though not significantly..”

Line 346 add “make to “and make clinical recommendations.”

Comments on the Quality of English Language

Main concern is this

Discrepancy about the total number recruited, enrolled and completing the study.  They list 88 as the n, but then say in line 84, that 8 participants “dropped out”.  Is the final n 96 or 80 or something else?  Please correct throughout the tables and paper on this point.

Reviewer 3 Report

Comments and Suggestions for Authors

The purpose of this study is to evaluate and compare the effect of walking and walking combined with manual techniques / elastic resistance exercise on the forward shoulder posture evaluated by acromial distance in healthy elderly. This is very interesting idea to study effect of easy form activity to this kind disease. 

However, in my opinion the main problem, weakness in this study are Participants. Were they unhealthy, had shoulder pain, back pain?  What the Authors would like to treat? The participants were healthy aged 60-80 years old, able to walk without gait aids for at least 20 minutes? Does it have a sense? 

In this study, the authors aimed to examine the effect of regular walking in healthy elderly. Participants of experimental groups underwent training sessions 2x/week for total of 4 weeks. 

What kind of stimulation we need for so spectacular effect after 8 meetings? I conclude the it couldn’t work. This is the result I observe in thit study. 

In discussion is too much introduction, information about results other scientists and too less explanations, interpretations own results.   

In the conclusion the authors write that “Results have showed that after the application of targeted walking training in the senior population for 60 minutes twice a week in combination with resistance exercises using elastic resistance on the trunk and upper limbs, there was a significant improvement in the position of the arm girdle on the left (non-dominant) upper limb. Influence of walking and its combination with manual techniques indicated, however not significantly, that these interventions may, when properly dosed, also improve shoulder posture in elderly people”.

I think that idea of this experiment was good but too less participants, too less training/therapy sessions and most important too positive conclusions.

These conclusions do not corelate with these results.

Round 2

Reviewer 1 Report

Comments and Suggestions for Authors

The content I reviewed has been well revised.